# A Cardiopulmonary Exercise Testing for Prescribing High-Intensity Interval Training Sessions with Elastic Resistance

**DOI:** 10.3390/ijerph20237097

**Published:** 2023-11-22

**Authors:** Lorena Flores Duarte, Victor Hugo Gasparini-Neto, Letícia Nascimento Santos Neves, Lenice Brum Nunes, Richard Diego Leite, Nuno Manoel Frade de Sousa, Luciana Carletti

**Affiliations:** 1Laboratory of Exercise Physiology (LAFEX), Physical Education and Sports Center, Federal University of Espírito Santo (CEFD-UFES), Vitória 29075-910, ES, Brazil; victorgasparini@gmail.com (V.H.G.-N.); leticiasantosneves@hotmail.com (L.N.S.N.); lenice.nunes@edu.ufes.br (L.B.N.); richard.leite@ufes.br (R.D.L.); luciana.carletti@ufes.br (L.C.); 2Center for Research in Sport and Physical Activity (CIDAF), Faculty of Sport Sciences and Physical Education, University of Coimbra (FCDEF-UC), 3040-256 Coimbra, Portugal; nuno.frade@gmail.com

**Keywords:** cardiopulmonary exercise testing, high-intensity interval training, oxygen consumption

## Abstract

This study aims to analyze the agreement of cardiopulmonary variables between a cardiopulmonary exercise test with elastic resistance (CPxEL) and high-intensity interval exercise with elastic resistance (EL-HIIE). Methods: Twenty-two physically independent participants were recruited. Visit one consisted of conducting a health survey and anthropometric assessment. On visit two, the participants performed CPxEL. After seven days, on visit three, the participants performed EL-HIIE. The CPxEL was carried out on a rubber mat demarcated by lines representing eight stages. The test consisted of alternating back and forth steps against elastic resistance. The increments were performed at a rate of one stage per minute, following a cadence controlled by a metronome calibrated by beats per minute (bpm). The EL-HIIE was performed at the stage corresponding to an intensity of ~85% VO_2max_, as determined by CPxEL. The EL-HIIE consisted of 10 × 1 min (work):1 min (passive rest), with a cadence of 200 bpm. Cardiopulmonary parameters, heart rate (HR), and oxygen consumption (VO_2_) were measured during exercise. Bland–Altman was applied to analyze the agreement between the HR and VO_2_ found in EL-HIIE and the values prescribed by CPxEL (~85–90% VO_2max_). Results: The HRpeak and VO_2_peak in the EL-HIIE showed good agreement with the VO_2CPxEL_ and HR_CPxEL_ values, showing an average difference of (−1.7 mL·kg^−1^·min^−1^) and (0.3 bpm). Conclusions: The results of the present study demonstrate the agreement of cardiopulmonary variables between the CPxEL and the EL-HIIE. Therefore, for a more specific prescription of EL-HIIE intensity, CPxEL can be used.

## 1. Introduction

High-intensity interval training (HIIT) was initially developed for middle- and long-distance (track) runners with the aim of training at intensities close to competition speed and enhancing aerobic and anaerobic fitness [1]. High-intensity interval training (HIIT) is defined as ‘near maximal’ efforts generally performed at an intensity that elicits at least 80% of the maximum oxygen consumption (VO_2max_) or 90% of the maximum heart rate (HR_max_), separated by periods of passive or active rest [2,3]. There is a growing consensus concerning the metabolic responses and physiological adaptations to HIIT. For instance, increased activity of mitochondrial enzymes is evident, and there is enhanced fat oxidation in the skeletal muscle [4]. As a result, this training modality has gained popularity, also reaching individuals who exercise for health purposes and seek an activity with a good time–efficiency ratio.

There are HIIT protocols that manipulate the effort–pause relationship and that are performed in different formats (e.g., running in the sand, jumping, and body work) to facilitate the involvement in exercise, as opposed to the restrictions of the traditional ergometers, for example, on treadmills and bicycles [5,6]. Furthermore, a recent study used elastic resistance in HIIT and demonstrated that this modality can be potentially favorable for the development of cardiorespiratory fitness [7]. Although elastic resistance can be used to perform HIIT, the ideal dimensions for prescribing HIIT using elastic resistance (i.e., prescription of intensity) remain unknown. In order to understand the prescription of HIIT using elastic resistance (EL-HIIE), it is necessary to better understand the acute cardiopulmonary responses to this exercise modality.

Researchers point out that rest intervals, intensity, and duration of work are variables that can be manipulated to prescribe different HIIT sessions [8,9]. Intensity can be controlled and individualized by specific running speed, percentage of HR_max_, percentage of VO_2max_, and rate of perceived exertion scale (RPE) [8]. However, cardiopulmonary indicators should result from a cardiopulmonary exercise test (CPx) performed on a treadmill or cycle ergometer due to their objectivity, precision, and effectiveness in determining these physiological parameters for exercise prescription [8]. The CPx is a gold-standard method that allows the determination of maximum physiological parameters (e.g., VO_2max_, HR_max_, maximum velocity, vVO_2max_, and respiratory exchange ratio) and submaximal (e.g., ventilatory thresholds one and two) which are used in the determination of exercise intensity.

The VO_2max_ is dependent on the ergometer used due to the specificity of the movement [10,11]; in that regard, for a more accurate prescription of the intensity in the EL-HIIE, the cardiopulmonary test of exercise with elastic resistance can be used (CPxEL). The CPxEL has good reproducibility in evaluating maximal and submaximal cardiopulmonary parameters [12]; however, it is important to check whether the cardiopulmonary parameters of the prescription are really equivalent to those found in the EL-HIIE session. Therefore, this study aims to evaluate the agreement between the cardiopulmonary parameters of the exercise session and the prescription parameters, based on a new proposal for a maximum incremental test, the CPxEL.

## 2. Materials and Methods

### 2.1. Subjects

The sample size was calculated using the G Power software program (version 3.1.4), by which a total number of 19 individuals was suggested, with statistical power (1-β) of 0.95 for the t test between two dependent means (corresponding pair), with an α-type error of 0.05 and a large effect size of 0.8. Twenty-two participants, including ten women and twelve men, participated in all experimental procedures (Table 1). The procedures were approved by the Human Research Ethics Committee of the Federal University of Espírito Santo (CAAE 09109319200005542). The participants were required to read and sign an online informed-consent form containing all information about the study procedures, risks, and benefits. This study adopted as inclusion criteria a BMI (≥18 and ≤25 kg·m^−2^), an age between 18 and 35 years, and physical independence (physical activity ≥150 min/week). Participants were excluded when there was cardiometabolic disease, use of dietary supplements or anabolic steroids, or suspected respiratory tract infections (for example, COVID-19). Participant characteristics are presented in Table 1 and Table 2.

### 2.2. Study Design

Participants visited the laboratory three times, with each visit separated by an interval of seven days, during morning hours (7:00 a.m. and 10:30 a.m.). Medical and anthropometric assessment were performed on the first visit. On the second visit, all participants performed the CPxEL until exhaustion to assess the maximal oxygen consumption and prescribed intensity of EL-HIIE. On the third visit, participants performed the EL-HIIE session (Figure 1).

### 2.3. Cardiopulmonary Exercise Test with Elastic Resistance (CPxEL)

The CPxEL followed the protocol Gasparini Neto et al. (2022) validated; additional details are available in that source [12]. The CPxEL was carried out to evaluate the maximum parameters and define the stage for the performance of the EL-HIIE. The test consisted of performing alternating steps back and forth against elastic resistance (Thera-band^®^ Tubing, Kuala Lumpur, Malaysia). The elastic resistance was attached to a polyamide military tactical belt (5 cm × 140 cm) inserted at the hip. Furthermore, the elastic resistance was attached to a load cell for force monitoring (200 kg; EMG System of Brazil, SP Ltd., Sao Paulo, Brazil). Force signals were collected by software (EMG Lab, version 1.03) at a sampling frequency of 1000 Hz. Data were analyzed using MatLab (MatLab; R2015a^®^, MathWorks, Natick, MA, USA). The results of the strength values were represented by the average strength in kilograms (kg) in the last 30 s of each stage.

Initially, the participants performed a brief familiarization with the protocol. Then, the silicone face mask for gas collection and the T31 coded™ heart rate sensor (Polar Electro Oy, Kempele, Finland) were adjusted. The movements in the CPxEL stages were performed until the participant reached the stage that corresponded to his maximum. A 3-min warm-up was performed, from line zero to line 2, at a cadence of 180 bpm (beats per minute). Afterward, the protocol consisted of increments of 1 stage per minute, following a cadence of 200 bpm on an 8-stage rubber mat (Figure 2). Participants were encouraged to follow a rhythm of 180 bpm (~90 steps/min) during the warm-up and 200 bpm (~100 steps/min) during the stages, following sounds emitted by a metronome app (Cifraclub^®^, Belo Horizonte, Brazil) plugged into a speaker. Constant verbal encouragement was applied to maintain the rhythm during stages. If the participant reached the last stage, an increase of 10 (ten) bpm was incrementally added every minute until exhaustion.

The ventilatory variables, oxygen consumption (VO_2_), and carbon dioxide output (VCO_2_) were collected using a metabolic gas analyzer (model: Cortex Metamax 3B, Germany), and measured breath-by-breath and analyzed at 20 s averages by Metasoft™. Heart rate was monitored continuously and collected using the T31 coded™ heart-rate sensor (Polar Electro Oy, Kempele, Finland). Afterward, 20 s averages were extracted using the Metasoft program.

The criteria for defining the maximum test were voluntary exhaustion, reaching at least 90% of the maximum heart rate predicted by the formula (220-age), RER above 1.05, or BORG-CR10 rate of perceived exertion scale (RPE) value at the ‘very difficult’ intensity (7 onwards) [13].

The ventilatory threshold 2 (VT2) was determined to verify how close the EL-HIIE session was to this intensity. Visual criteria were used, based on the response of ventilatory equivalents (VE/VO_2_ and VE/VCO_2_) and carbon dioxide tension (P_ET_CO_2_). Three evaluators independently and blindly evaluated the results, and the study employs the limit of agreement of at least two evaluators (ICC, 0.93). Identification of VT2 was based on the following criterion: the moment of the lowest point of the VE/VCO_2_ with subsequent elevation beyond the moment of the gradual decline of the end-tidal carbon dioxide tension (P_ET_CO_2_) was considered [14].

### 2.4. High-Intensity Interval Exercise with Elastic Resistance (EL-HIIE)

The session was prescribed at the stage corresponding to an intensity of 85–90% of the VO_2max_, as determined by the CPxEL. In EL-HIIE, the following were performed: 10 × 1 min (work):1 min (passive rest) (60 s protocol—adapted from Little et al. (2011)). The series consisted of alternating forward and backward movements against elastic resistance (Theraband^®^ Tubing, Malaysia) at a cadence of 200 bpm, monitored by a metronome app (Cifraclub^®^, Brazil). Participants, as in the CPxEL, used the belt attached to the elastic tube (Silver tube, Thera-band^®^ Tubing, Malaysia), the silicone face mask, and the T31 coded™ heart rate sensor (Polar Electro Oy, Kempele, Finland). Ventilatory variables (VO_2_ and VCO_2_) were collected using a metabolic gas analyzer (model Cortex Metamax 3B, Weil am Rhein, Germany) and measured breath-by-breath. Afterward, VO_2peak_ (average of the last 10 s for each bout) and VO_2average_ (average of the 60 s for each bout) were calculated.

Heart rate was monitored continuously by the T31 coded™ heart rate sensor (Polar Electro Oy, Kempele, Finland). Then, the HR_peak_ (average of the last 10 s for each bout) and HR_average_ (average of the 60 s for each bout) were calculated.

### 2.5. Statistical Analysis

Data were expressed as mean and standard deviation (SD) and analyzed for normality (Shapiro–Wilk test) using the GraphPad Prism 9 software. To compare and evaluate the agreement between the cardiopulmonary responses (VO_2_ and HR) elicited by EL-HIIE and those prescribed by CPxEL, we employed the paired *t*-test and Bland–Altman analysis. One-way repeated-measures ANOVA and Tukey’s multiple comparison tests were employed to ascertain the distinctions between the peak and average values of VO_2_ and HR during EL-HIIE, compared with the corresponding values obtained from the prescription provided by CPxEL. The *p* values were exact, and statistical significance was defined as *p* < 0.05. Cohen’s d effect size from an arbitrary scale was calculated and classified as trivial (0–0.19), small (0.20–0.49), moderate (0.50–0.79), or large (≥0.8) to determine the magnitude of differences [15].

## 3. Results

### 3.1. CPxEL vs. EL-HIIE (Peak Values)

Prescription values for the CPxEL and peak values for the EL-HIIE are shown in Table 3. The average stage performed during the EL-HIIE was stage 3, with an external load applied through elastic force averaging 66.6 ± 3.2% of the maximum load determined in the CPxEL. There was a statistical difference between the VO_2CPxEL_ and the VO_2session_, with a small effect size (ES). The VO_2session_ was 5% higher than the VO_2CPxEL_. Over the 10 min bout in the EL-HIIE, VO_2_ remained at 147.6 s (2.46 min) in the prescription target range (≥85% VO_2max_), in the following proportions: bout 1, 5.4 s; bout 2, 14.0 s; bout 3, 15.4 s; bout 4, 18.1 s; bout 5, 15.5 s; bout 6, 15.9 s; bout 7, 15.4 s; bout 8, 15.4 s; bout 9, 16.3 s; and bout 10, 17.2 s.

There was no statistical difference between the HR value of the prescription and that of the EL-HIIE. Over the 10 min bout in the EL-HIIE, HR remained at 125.3 s (2.08 min) in the prescription target range (≥85% VO_2max_), in the following proportions: bout 1, 2.2 s; bout 2, 5.4 s; bout 3, 10.0 s; bout 4, 12.7 s; bout 5, 14.3 s; bout 6, 15.9 s; bout 7, 17.7 s; bout 8, 18.6 s; bout 9, 19.0 s; and bout 10, 22.2 s.

In EL-HIIE, two participants reached a VO_2_ of VT2 as determined by CPxEL, six were 5.7% above VT2, and fourteen were 8.3% below VT2. In addition, one participant reached the HR of VT2 determined by CPxEL, seventeen were 6.8% below VT2, and four were 1.7% above VT2.

### 3.2. CPxEL vs. EL-HIIE (Average Values)

Prescription values of CpxEL and average values of 60 s EL-HIIE are shown in Table 4. The VO_2_ of the EL-HIIE average value was 30.4 ± 4.4 mL·kg^−1^·min^−1^, 74.8 ± 15.3% of the VO_2max_. The VO_2session_ was 15% less than the VO_2CpxEL_. There was a propensity of VO_2_ stabilization from bout 3 onward.

The average HR of the session was 157 ± 16 bpm, 84 ± 53.7% of HR_max_. The HR_session_ was 8.4% less than the HR_CpxEL_. From bout 3, the average HR of the session showed a percentage of 82.3% of HR_max_, increasing at the end of bout 10 to 89% of HR_max_.

### 3.3. Bland–Altman Analysis

The mean (peak) difference between VO_2session_ and VO_2CpxEL_ was −1.7 mL·kg^−1^·min^−1^ with concordance limits of +4.8 to −8.3 mL·kg^−1^. The mean (average) difference between VO_2CpxEL_ and VO_2session_ was 5.4 mL·kg^−1^·min^−1^ with concordance limits of +11.8 to −0.9 mL·kg^−1^·min^−1^ (Figure 3).

The mean (peak) difference between HR_CpxEL_ and HR_session_ was 0.3 bpm with concordance limits of +16.3 to −16.9 bpm. The mean (average) difference between HR_CpxEL_ and HR_session_ was 14.4 bpm with concordance limits of +33.1 to −4.1 bpm (Figure 3).

### 3.4. Difference between Prescription Parameters and Exercise Session Parameters

The VO_2CpxEL_ represented 88.1% of VO_2max_, the VO_2session_ (peak), 92.6% of VO_2max_, and the VO_2session_ (average), 74.8% of VO_2max_. The distinction between the VO_2CPxEL_ and the VO_2session_ (peak) was −1.7 mL·kg^−1^·min^−1^, *p* = 0.47. The distinction between VO_2CPxEL_ and VO_2session_ (average) was 5.4 mL·kg^−1^·min^−1^, *p* = 0.00. The distinction between VO_2session_ (peak) and VO_2session_ (average) was 7.1 mL·kg^−1^·min^−1^, *p* = 0.00 (Figure 4).

The HR_CPxEL_ represented 91.9% of HR_max_, the HR_session_ (peak), 92.1% of HR_max_, and HR_session_ (average), 78.4% of HR_max_. The distinction between HR_CPxEL_ and HR_session_ (peak) was −0.3 bpm, *p* = 0.99. The distinction between HR_CPxEL_ and HR_session_ (average) was 14.4 bpm, *p* = 0.005. The distinction between HR_session_ (peak) and HR_session_ (average) was 14.7 bpm, *p* = 0.00 (Figure 5).

## 4. Discussion

The present study aimed to assess the agreement between the cardiopulmonary variables of the EL-HIIE and the values used in a prescription based on the CPxEL. The significance of movement specificity in prescribing exercise intensity is well-recognized, as the measurement of VO_2_ relies on factors such as the type of equipment (treadmill, cycle ergometer), the protocol employed, and the nature of effort exerted during the testing [10].

Our main findings demonstrated a good agreement between the cardiopulmonary parameters of the EL-HIIE and the prescription parameters determined by the CPxEL. CPxEL was initially proposed to determine VO_2max_ and ventilatory thresholds in healthy individuals, and our results highlight the importance of its use in prescribing correctly a 10 × 1:1 min protocol in the EL-HIIE at an intensity of 88.1 ± 5.8% of VO_2max_.

This study defined the exercise intensity prescription parameter according to the reference already used in the literature (≥85% of VO_2max_) [16,17]. In these previously mentioned studies, the individuals exercised within a workload corresponding to the submaximal fraction of the VO_2max_, while the HR, lactate production, speed, power, and classification of the perceived exertion during the execution of the exercise were monitored. Given the few data sources available in the literature, our study directly monitored VO_2_ in EL-HIIE. Therefore, it was possible to confirm the good agreement between the VO_2session_ (peak) of EL-HIIE and the VO_2_ prescribed by CPxEL.

The present study, in addition, monitored HR during EL-HIIE. In sports practice, the use of this metric is frequent, although the literature points out that there may be limitations, mainly due to the delay in changes in the heart rate at the beginning of the exercise [18]. Our findings demonstrated good agreement between the HR_session_ (peak) of the exercise session and the HR CPxEL prescription. Therefore, this evidence points out that monitoring HR_session_ (peak) in EL-HIIE ensured that participants were at an intensity equivalent to the specific test prescription (CPxEL).

Although the main focus of prescribing and monitoring in the EL-HIIE was not the external load, it was possible to identify the average load used during the exercise execution (66.6% of the peak force reached in the CPxEL), as continuously measured via a load cell. The load used in the EL-HIIE is similar to the prescription parameters established in the literature. In the study by Hood et al. (2011), a HIIT exercise session was performed on a cycle ergometer, prescribed at ∼60% of the peak power reached during the VO_2peak_ test. The bouts in this study provoked a cardiopulmonary effort corresponding to ∼80% to 95% of HR reserve [19]. The cardiopulmonary results found in EL-HIIE (∼84% to 92% HR_max_), from 66.6% of peak force, confirm that the exercise protocol of our study was performed at high intensity.

Our findings, moreover, indicated lower values of HR and VO_2_ in the average values compared to the peak values. This result was expected, since lower average values of HR and VO_2_ can be attributed to the passive recovery interval of our protocol. The present study agrees with a previous study that observed higher average values of HR and VO_2_ in active intervals in HIIT with an effort–pause ratio of 1:1 [20].

In EL-HIIE, participants remained for approximately 24.6% of the total exercise time within the prescription target zone (≥85% VO_2max_). Comparing our results with the literature is difficult, since the intensity and duration of HIIT exercise differ significantly. In addition, few studies show how long participants remain in the prescription target zone. Furthermore, to the best of our knowledge, this was the first study to investigate the agreement between the CPxEL prescription’s cardiopulmonary parameters and the EL-HIIE session’s parameters. On the other hand, future research with EL-HIIE can manipulate the duration of bouts and recovery intervals, such as the high-intensity decreasing interval training (HIDIT) protocol performed by cyclists [21]. This protocol is characterized by bout intervals varying between long and short, starting with 3 min and ending with 30 s. The HIDIT performed in cyclists showed a longer time of high-intensity exercise, which could be a strategy for EL-HIIE. The 60 s:60 s protocol performed in the present study was a strategy to ensure an exercise intervention capable of being performed by healthy individuals and clinical populations [22].

Some limitations need to be pointed out: our study evaluated only healthy and eutrophic young individuals, and these findings cannot represent the behavior of subjects with different levels of physical conditioning, like obese and sedentary individuals. Another limitation was the mixed sample (men and women), because men and women can have up to a 30% difference for VO_2max_, but another side of the question is that a mixed sample increases the ecological validity of results. Despite these limitations, it is essential to highlight that our proposal was safe and presented good agreement between CPxEL and EL-HIIE. Thus, we encourage studies to apply our protocol to different populations and on a large scale to understand the better application of these findings in daily life and in longitudinal exercise prescription of physical exercise in different intensity domains.

Although it is not the primary outcome of this research, there is interest in the scientific literature on multicomponent training modalities. This modality offers a combination of physical capabilities (muscle strength, cardiorespiratory endurance, balance, and flexibility) in the same exercise session [23,24]. Considering that EL-HIIE can be a potential stimulus for developing cardiorespiratory fitness and lower-limb strength in young, active individuals [7], future research may investigate whether EL-HIIE can be a multicomponent exercise strategy applied in different populations.

## 5. Conclusions

The study shows good agreement between CPxEL and EL-HIIE for cardiopulmonary variables (HR and VO_2_). These findings highlight the importance of specificity of movement for an adequate prescription of exercise intensity. For health and performance purposes, correctly prescribed exercise intensities allow for increased predictability of adaptive responses.

## Figures and Tables

**Figure 1 ijerph-20-07097-f001:**
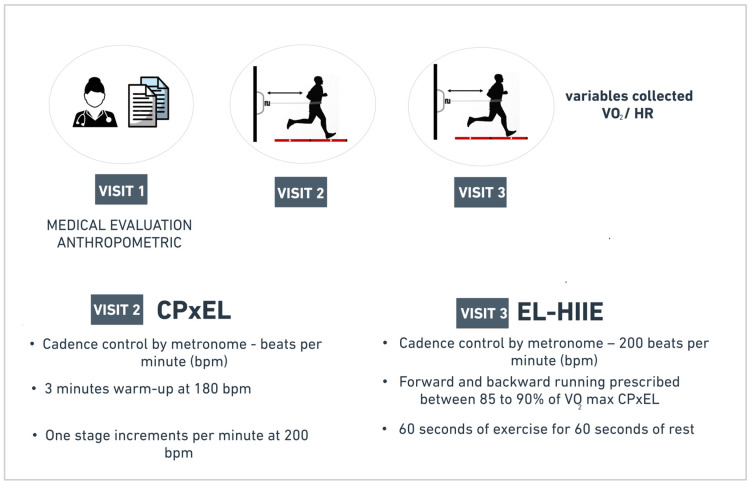
Study design. Day 1: Medical and anthropometric evaluation. Day 2: Cardiopulmonary test with elastic resistance (CPxEL). Day 3: High-intensity interval exercise session (EL-HIIE). VO_2_: oxygen consumption; HR: heart rate.

**Figure 2 ijerph-20-07097-f002:**
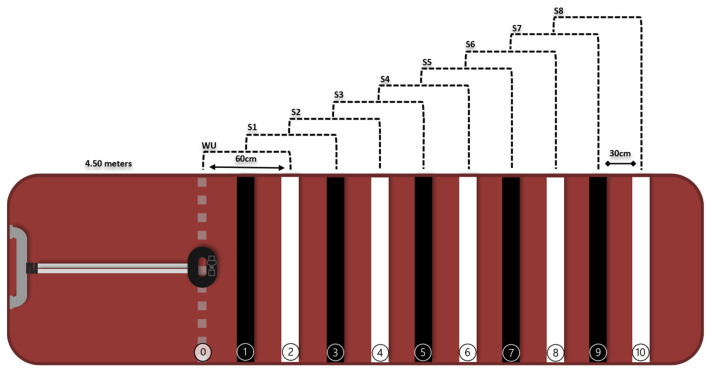
Schematic of the rubberized mat (length of 4.50 m) demarcated with 11 lines (0–10)—30 cm between lines. S0 (WU) and 8 (eight) stages (S1 to S8)—60 cm between stages, interspersed with black and white colors. WU: warm-up. Source: [12].

**Figure 3 ijerph-20-07097-f003:**
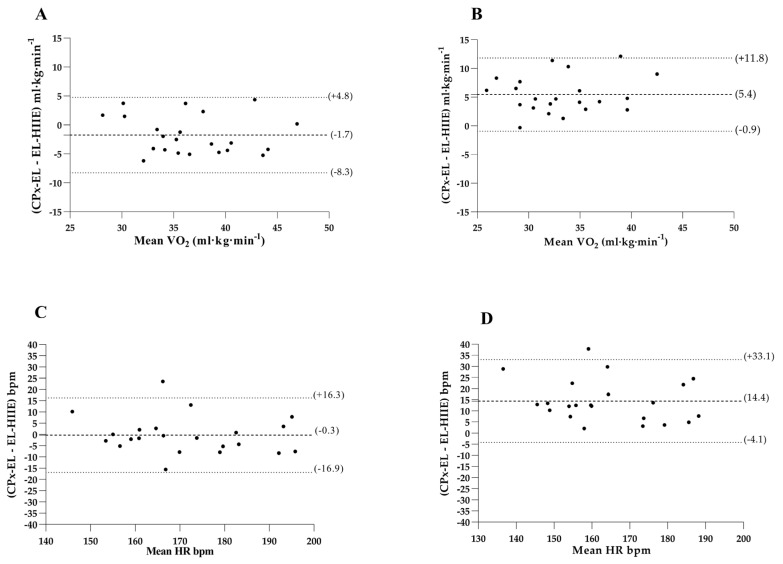
Bland–Altman plot: Y axis—upper generated line (indicates upper limit +2SD), a line drawn in the center (indicates the difference between means), and lower generated line (indicates lower limit -2SD). VO_2_: oxygen consumption; HR: heart rate. (**A**) Limits of agreement of the Bland–Altman technique between the VO_2CpxEL_ and VO_2session_ (peak). (**B**) Limits of agreement of the Bland–Altman technique between the VO_2CpxEL_ and the VO_2session_ (average). (**C**) Limits of agreement of the Bland–Altman technique between the HR_CpxEL_ and the HR_session_ (peak). (**D**) Limits of agreement of the Bland–Altman technique between the HR_CpxEL_ and the HR_session_ (average).

**Figure 4 ijerph-20-07097-f004:**
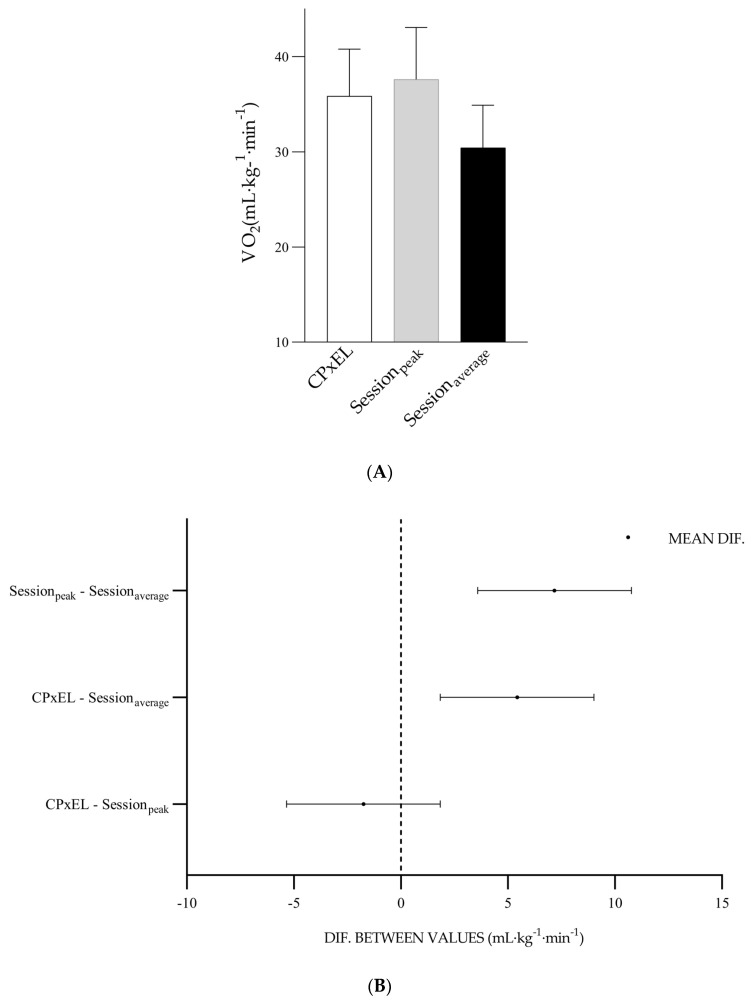
(**A**) Values and standard deviation of VO_2CPxEL_, VO_2session_ (peak) EL-HIIE, and VO_2session_ (average) EL-HIIE. (**B**) Graph of the difference between the mean values of VO_2CPxEL_, VO_2session_ (peak) EL-HIIE, and VO_2session_ (average) EL-HIIE. X-axis: difference between the values.

**Figure 5 ijerph-20-07097-f005:**
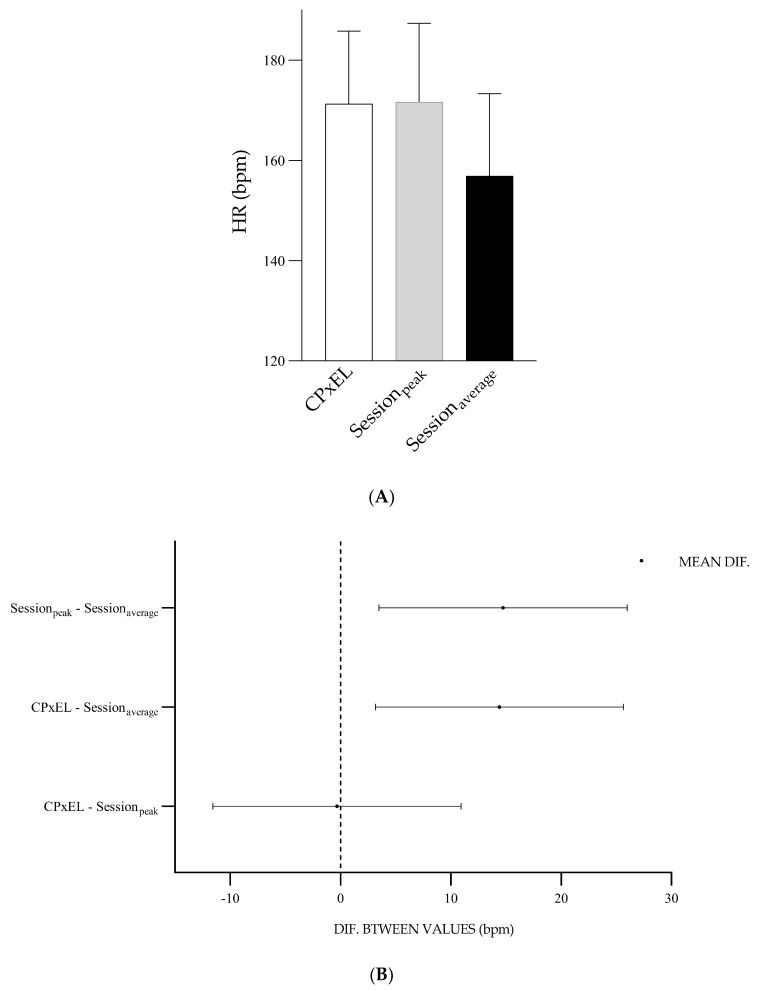
(**A**) Values and standard deviation of HR_CPxEL_, HR_session_ (peak), and HR_session_ (average). (**B**) Graph of the differences between the mean values of HR_CPxEL_, HR_session_ (peak)EL-HIIE, and HR_session_ (average). X-axis: the difference between the values.

**Table 1 ijerph-20-07097-t001:** Characteristics of the participants men (n = 12).

Age (years)	27.2 ± 3.7
Height (m)	1.76 ± 0.08
Body Mass (kg)	71.9 ± 10.2
BMI (kg·m^−2^)	23.1 ± 2.2
VO_2max_ (mL·kg^−1^·min^−1^)	43.7 ± 3.5
HR_max_ (bpm)	185 ± 12.2

Values are presented as the mean and standard deviation. BMI: body mass index; HR: heart rate; VO_2max_: maximal oxygen uptake.

**Table 2 ijerph-20-07097-t002:** Characteristics of the participants women (n = 10).

Age (years)	27.2 ± 5.0
Height (m)	1.63 ± 0.05
Body Mass (kg)	59.1 ± 5.2
BMI (kg·m^−2^)	22.1 ± 2.3
VO_2max_ (mL·kg^−1^·min^−1^)	36.5 ± 2.9
HR_max_ (bpm)	188 ± 8.6

Values are presented as the mean and standard deviation. BMI: body mass index; HR: heart rate; VO_2max_: maximal oxygen uptake.

**Table 3 ijerph-20-07097-t003:** CPxEL and EL-HIIE (peak values) (n = 22).

Physiological Parameters	CPxEL	EL-HIIE Peak	*p*	Cohen’s d
VO_2_ (mL·kg^−1^·min^−1^)	35.8 ± 4.9	37.6 ± 5.4	0.02	0.3 ^S^
%VO_2max_	88.1 ± 5.8	92.6 ± 3.8	-	-
HR_max_ (bpm)	171.3 ± 14.4	171.6 ± 15.7	0.86	0.0 ^T^
%HR_max_	91.9 ± 35	92.1 ± 48.1	-	-

CPxEL: cardiopulmonary test with elastic resistance; EL-HIIE: high-intensity interval exercise with elastic resistance; VO_2_: oxygen consumption; %VO_2max_: percentage of VO_2max_; HR: heart rate; %HR: percentage of maximum heart rate; S: small; T: trivial; Cohen’s d: effect size—trivial (0–0.19), small (0.20–0.49), moderate (0.50–0.79), and large (≥0.8). *p* < 0.05.

**Table 4 ijerph-20-07097-t004:** CpxEL vs. EL-HIIE (average values) (n = 22).

Physiological Parameters	CpxEL	EL-HIIE Average	*p*	Cohen’s d
VO_2_ (mL·kg^−1^·min^−1^)	35.8 ± 4.9	30.4 ± 4.4	0.00	1.1 ^L^
%VO_2max_	88.1 ± 5.8	74.8 ± 15.3	-	-
HR_max_ (bpm)	171.3 ± 14.4	156.9 ± 16.3	0.00	1.0 ^L^
%HR_max_	91.9 ± 35	84 ± 53.7	-	-

CpxEL: cardiopulmonary test with elastic resistance; EL-HIIE: high-intensity interval exercise with elastic resistance; VO_2_: oxygen consumption; %VO_2max_: percentage of VO_2max_; HR: heart rate; %HR: percentage of maximum heart rate; L: large; Cohen’s d: effect size—trivial (0–0.19), small (0.20–0.49), moderate (0.50–0.79), and large (≥0.8). *p* < 0.05.

## Data Availability

Data are contained within the article.

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
