# Peer review of "A Cardiopulmonary Exercise Testing for Prescribing High-Intensity Interval Training Sessions with Elastic Resistance"

_ijerph, 2023, doi:10.3390/ijerph20237097_

Round 1

Reviewer 1 Report

Comments and Suggestions for Authors

I wanted to congratulate the authors for the work they have done. This is a high-quality manuscript that merits publication. The introduction explains well the background of the study. The methodology is well written to be able to reproduce the study. The results are presented clearly. The discussion is concise. I only recommend a minor revision with the following comments:

1. The authors have used the terms “visit 1, visit 2, visit 3” and “day 1, day 2, day 3” interchangeably. I recommend to use the visit number instead of the study day since visit 2 corresponds to day 8 of the study.

2. The sample size calculation should be described.

3. I recommend to add a paragraph to describe the limitations of the study.

4. For similar studies in the future, the authors could consider a crossover design, randomizing the order of interventions, to minimize bias.

Author Response

Dear reviewer

I would like to thank for your considerations on our article. We review all points. 

Reviewer 2 Report

Comments and Suggestions for Authors

Authors:

The abstract should provide a clear and structured overview of your study. It's currently a bit fragmented. Consider reorganizing it into distinct sections, such as background, methods, results, and implications.

 Introduction:

Start with a brief introduction that highlights the significance of the study. Explain why it's important to analyze the agreement between cardiopulmonary parameters in EL-HIIE and CPxEL prescription.

Results:

Very clear.

Conclusion:

Your conclusion is succinct and to the point, summarizing the key findings of your study and emphasizing the importance of test specificity in exercise prescription. However, you might consider expanding the conclusion slightly to provide more context or practical implications of the good agreement observed between the physiological parameters in CPx-EL and EL-HIIE.

Overall, your results section has been positively acknowledged and is considered highly clear. This is a significant advantage, as clear results aid readers in swiftly comprehending your research findings.

Author Response

(The authors gave the same response as above.)

Reviewer 3 Report

Comments and Suggestions for Authors

The purpose of this study is quite difficult to understand and there is a lack of a strong rationale to support it.

Below are my specific comments.

Line 18: Define CPxEL before abbreviating.

Line 19: “…participants performed…”

Lines 21-22: What was the step count per minute?

Line 22: What does the following mean “…metronome in an 8-stage rubber mat.”

Line 23: How can you exercise at a specific VO2 unless the participants were exercising with respiratory gases being analyzed? Do you mean exercising at an intensity/power output that is estimated to produce the targeted 85% VO2max?

Line 24: A cadence of 200 steps a minute – is this what you mean?

Lines 25-26: I have no idea what you are investigating in this study. Comparing EL-HIIE to CPxEL makes no sense. Please better describe what you did in this study.

Line 35: HIIT was originally designed for middle and long-distance runners–track athletes.

Lines 41-43: Please revise this sentence due to the poor grammar.

Lines 46-49: Very poorly written, ‘body weight’ is not an activity. This sentence needs to be revised.

Lines 56-58: But were the exercise modes a match for energy expenditure? You need to better critique the results of this study.

Lines 61-63: Elastic resistance combined with aerobic exercise is not commonly used.

Lines 85-88: I still have no idea what you are investigating. Are you validating an exercise test?

Line 92: How was the sample size determined to produce statistical power?

Table 1: Please separate male and female characteristics.

Lines 108: What do you mean by “anamnesis”?

Line 150: Define the ‘very difficult intensity’.

Figure 2: This makes no sense, are you suggesting that at stage 8 a 3-metre stride is required? This is impossible.

Lines 185-190: Each ‘bout’, do you mean ‘stage’ from the test?

Lines 221-224: What is ‘LV2’?

Line 229: Need to insert ‘max’ after ‘HR’.

Line 237: Do you mean ‘median’?

Table 3: The same data has been included as in Table 2. Please delete.

Lines 251-252: Please revise and make the title more concise.

Lines 273-275: Please revise and make the title more concise.

Line 281: ’91.9’

Line 282: I think ‘784’ percent is incorrect.

Figures 4 and 5. Please improve these figures. 4B and 5B are difficult to understand.

Line 322: “Prescribing intensity session’ – makes no sense because all exercise involves an ‘intensity’.

Line 369: “…protocol uses intervals….””

Lines 383-386: Please expand upon the Conclusion. 

Comments on the Quality of English Language

There are grammatical errors and poor structuring of sentences throughout the manuscript.

Please thoroughly read over the manuscript and improve the quality of your writing.

Author Response

Dear reviewer

I would like to thank for your considerations on our article. We review all points. I highlight:

1- line 237 of the previous version - these are average values.

2- Tables 2 and 3 of the previous version are different. The first peak values of the session, the second average values.

3- This link will help you understand CPx-EL in more detail: https://www.youtube.com/watch?v=tE3VEwm9Jww

Round 2

Reviewer 3 Report

Comments and Suggestions for Authors

I am satisfied with your response.

Comments on the Quality of English Language

English quality is satisfactory.